# The Role of Visceral Therapy in the Sexual Health of Women with Endometriosis during the COVID-19 Pandemic: A Literature Review

**DOI:** 10.3390/jcm11195825

**Published:** 2022-09-30

**Authors:** Tomasz Gozdziewicz, Grazyna Jarzabek-Bielecka, Dawid Luwanski, Malgorzata Wojcik, Katarzyna Plagens-Rotman, Malgorzata Mizgier, Magdalena Pisarska-Krawczyk, Witold Kedzia

**Affiliations:** 1Division of Gynecology, Department of Perinatology and Gynecology, Poznan University of Medical Sciences, 61-758 Poznan, Poland; 2Center for Sexology and Pediatric, Adolescent Gynecology, Division of Gynecology, Department of Perinatology and Gynecology, Poznan University of Medical Sciences, 61-758 Poznan, Poland; 3Department of Physiotherapy, Faculty of Physical Culture in Gorzów Wielkopolski, Poznań University of Physical Education, 61-871 Poznan, Poland; 4Department of Sports Dietetics, Poznan University of Physical Education, 61-871 Poznan, Poland

**Keywords:** endometriosis, pain, visceral therapy, sexual health

## Abstract

Patients with endometriosis had limited possibilities for contemporary diagnosis and treatment during the SARS-CoV-2 (COVID-19) pandemic. Surgeries that may have eliminated pain or restored fertility were postponed. Endometriosis may affect the vagina, peritoneum, bladder, or other organs outside the pelvis and impact women’s sexual health, especially during pandemics. Holistic care of patients is crucial to improving their lives and sexual health. A scoping review was conducted to analyze the relevant literature in light of our experience in gynecology and physiotherapy during the COVID-19 pandemic.

## 1. Introduction

Patients with endometriosis had limited access to diagnosis and treatment after the beginning of the COVID-19 pandemic. Outpatient medical specialist visits and surgeries were postponed due to fear of contracting COVID-19 (caused by the SARS-CoV-2 virus). The need for isolation to avoid disease spread aggravated the problem, which exerted negative effects on women’s sex lives and psychological health. Endometriosis is often connected with painful intercourse or dyspareunia.

Endometriosis is characterized by the presence of endometrial implants outside the uterine cavity. The most common locations are ovaries, tubes, uterine ligaments, and the pelvic peritoneum. However, implants that also affect the vagina, bladder, rectum, or other organs outside the pelvis are referred to as deep infiltrating endometriosis (DIE). Endometrial implants bleed during menstruation, often causing severe pain during menstruation, intercourse, defecation, or urination. Severe menstrual bleeding connected with acute pelvic and spinal pain are common sequalae of this disease.

Endometriosis affects approximately 11% of females, including over 1 million in Poland. The mean time to diagnosis, beginning at first symptoms, exceeds 10 years. Hormones that regulate menstruation similarly affect the ectopic endometrium. Chronic pelvic inflammation and bleeding from endometrial implants can produce clinical symptoms, pelvis adhesions, ovarian cysts, and other pathological conditions.

## 2. Etiology of Endometriosis

Endometriosis is an enigmatic disease. Although its etiopathology remains unknown, there are several popular theories including retrograde menstruation (Sampson 1927), metaplasia (Waldeyer 1890), and induction (Levande and Norman 1955). New insights into etiology focus on abnormal fetal programming due to preeclampsia, prematurity, or maternal cigarette smoking during pregnancy. Heavy menstrual blood flow is associated with early menarche, prolonged menstruation, or even short cycles. All lead to clinical symptoms; however, the cyclical nature of endometriosis-associated pain is its most characteristic feature.

## 3. Diagnosis of Endometriosis

Clinical symptoms together with careful clinical examination is crucial to the preliminary diagnosis of endometriosis. During speculum examination, blue implants or painful nodules may be observed in the vagina fornix, in addition to areas of thickening or shortened uterosacral ligaments. A retroflexed uterus also may suggest endometriosis [1]. Lack of uterine mobility during bimanual examination may suggest peritoneal endometriosis and adhesions [2]. Anterior vaginal wall tenderness during physical exam is another typical symptom in women with endometriosis [3]. Rectovaginal digital examination may reveal deep endometriosis involving the rectosigmoid colon.

Imaging tools assist with noninvasive diagnoses and include transvaginal ultrasound or magnetic resonance. Endometriosis imaging protocols often involve application of gel into the vagina and gel or water into the rectum [4].

Certain medical history factors may increase the risk of endometriosis. These include: (1) a familial history, particularly among first degree relatives (which conveys a 6–7 fold higher risk of endometriosis) [5]; (2) prematurity [6]; (3) low birth weight and abnormal uterine bleeding during the neonatal period [7]; (4) formula feeding of newborns [6]; (5) reduced growth during childhood [8]; (6) childhood abuse [9]; (7) painful menarche affecting social life; (8) pain-related school absences; (9) unsatisfactory response to non-steroidal anti-inflammatory drugs (NSAIDs) [10]; (10) migraines [11]; (11) low body mass index (BMI); (12) pigmented skin lesions; (13) freckles [12]; (14) infertility [13]; (15) cyclic pain that increases during menstruation [14]; (16) pain during menstruation from digestive or urological systems, diaphragm, lungs, or sciatica [15]; (16) fatigue syndrome: pain, insomnia, depression, and stress at work [16]; (17) obstetric history: miscarriages, adverse pregnancy outcomes [17,18]; (18) pelvic surgery for endometriosis or other gynecological indications [19,20]; and (19) autoimmune diseases [21]. However, to date no study has assessed whether using questionnaires or symptom diaries shortens or improves the diagnosis of endometriosis for screening or triaging symptomatic patients compared to traditional history-taking techniques.

Prescription of hormonal drugs such as progestins or oral contraceptives may be used as a clinical test in certain situations. However, such management can be proposed only for patients not planning to be pregnant soon. During the COVID-19 pandemic, patients underwent virtual or *telehealth* evaluations. Analysis of the patient’s medical history and symptoms can facilitate diagnosis of endometriosis and implementation of medical treatment. Some sequalae of endometriosis―such as ovarian tumors with suspected malignancy, bowel obstruction due to DIE, and hydronephrosis due to DIE ureter occlusion―indicate the need for emergent surgery. Other endometriosis-related surgeries were routinely postponed during the pandemic. Consequently, patients sought other solutions for managing pain. In addition to oral contraceptives and progestins, many patients were administered GnRH analogs, danazol, or aromatase inhibitors. GnRH agonists include Goserelin (Zoladex^®^) and Tryptorelin (Decapeptyl^®^); GnRH antagonists include injections of Degarelix (Firmagon^®^), Abarelix (Plenaxis^®^), Cetrorelix (Cetrotide^®^), and Ganirelix (Antagon^®^/Orgalutran^®^), as well as oral drugs such as Linzagolix (Yselty^®^, still not registered), Relugolix (Ryeqo^®^/Orgovyx^®^), and Elagolix (Orlissa^®^).

Although the gold standard for endometriosis diagnosis in the past was direct visualization of endometrial implants, a negative laparoscopy did not necessarily exclude this disease from consideration. Nowadays, the European Society of Human Reproduction and Embryology recommends consideration of endometriosis in patients presenting with the following cyclical and non-cyclical signs and symptoms: dysmenorrhea, deep dyspareunia, dysuria, painful bowel movements or *dyschezia*, painful rectal bleeding or *hematuria*, shoulder tip pain, lung collapse related to menstruation or *catamenial pneumothorax*, cyclical cough/hemoptysis/chest pain, cyclical scar swelling and pain, fatigue, or infertility. Moreover, advances in the quality and availability of imaging modalities for at least some forms of endometriosis may reliably detect or exclude endometriosis [22].

## 4. Dyspareunia and Sexual Health

Dyspareunia is the most common sexual health symptom of endometriosis. It is considered a form of female sexual dysfunction (FSD) and can take various forms, including decreased sexual desire, sexual aversion, disturbances in the course of excitement, orgasmic disorders, dyspareunia, and fear of vaginal penetration or *vaginismus*. FSD is common and can involve one, several, or all stages of sexual reaction (i.e., desire, excitement, orgasm, and relaxation).

While it is normal to experience pleasure and satisfaction during sexual intercourse, many patients report experiencing pain, humiliation, fear of sexual abuse, or objective treatment. Pain is sometimes a barrier to achieving complete sexual satisfaction. This pain may be acute, chronic, or recur in the genitals or small pelvis before, during, or after sexual intercourse. Experiencing pain at the beginning of intercourse may condition the individual to expect pain with each subsequent sexual contact. Usually, the pain increases with the intensity and duration of intercourse. This pain often reduces pleasure during the lead up to sex. When sex is associated with pain there is an increase in unpleasant sensations and a decrease in pleasurable sensations.

Sexual pain, at any age, can lead an individual to avoid sexual contact and may affect the sex partner. Some partners may feel comfortable forming non-sexual relationships, but experience fear and aversion when asked to assume the role of lover. Sex partners can become conditioned to avoid any activity temporally related to painful intercourse.

Dyspareunia refers to genital pain before, during, or immediately after intercourse. Although this condition can affect both sexes, it is much more common in females. Cross-sectional epidemiological studies by Danielsson et al., which focused on sexually active Swedish females, found a prevalence of 13%. Dyspareunia was reported twice as frequently by young females (20–29 years old) compared to older (50–60 years old) females. Dyspareunia involves pain associated with sexual intercourse, without shrinkage of the vulva and vagina, and is different from vaginismus, which prevents penetration.

Dyspareunia can be classified according to its physical location. (1) Superficial (shallow) dyspareunia is localized to the vestibule of the vagina; (2) deep dyspareunia involves the vaginal vault; and (3) generalized dyspareunia encompasses the entire vagina. Dyspareunia can also be characterized according to chronology. (1) Primary dyspareunia appears at the first sexual contact, whereas (2) secondary dyspareunia occurs as a result of some other activity (which can sometimes be revealed by a well-conducted interview). Dyspareunia can also be classified according to its relationship to time during intercourse. (1) Early dyspareunia manifests at the beginning of intercourse and disappears after its completion, whereas (2) late dyspareunia occurs at the end of intercourse, or even a few hours later.

Dyspareunia can be continuous, manifesting during every occurrence of intercourse, or intermittent, occurring only during certain positions or during specific menstrual cycle phases (usually during ovulation, especially in women with PMS symptoms). Dyspareunia may also vary relative to the sexual partner. The psychological and interpersonal effects of this disorder are critically important considerations.

Medical histories of dyspareunia patients should be taken with great delicacy and empathy. At the same time, accurate recording of a comprehensive medical history is important for defining the nature of dyspareunia. Although its etiopathogenesis may be multifactorial, in many females dyspareunia relates to organic and/or psychogenic dysfunction. Potential psychogenic etiologies of dyspareunia include childhood sexual abuse, excessive shame or guilt during sexual intercourse, fear of intercourse (especially loss of virginity or *defloration*), and difficulties achieving sexual readiness. The primary somatic causes of dyspareunia include reproductive organ inflammation, vaginismus, and labia hyperaesthesia. Endometriosis is a common organic cause of dyspareunia, which can also include pelvic pain syndrome and dysmenorrhea.

Among the various the diseases associated with dyspareunia, endometriosis is one of the most difficult to diagnose. Endometriosis occurs in approximately 10% of women and is often associated with dyspareunia. Eskanazi et al. showed that the positive prognostic value for dyspareunia in the diagnosis of endometriosis is 40%. The first attempt to assess selected elements of sexual quality of life in females with endometriosis was carried out by Ferraro et al. They showed that the strongest dyspareunia was reported by women with endometriosis foci within the sacro-uterine ligaments.

Presently, we are unable to comprehensively explain the mechanisms of pain development in patients with endometriosis. Adhesions are a possible cause, as they often accompany endometriosis and may impede physiological reproductive organ changes during sexual intercourse. Inflammatory mediators found in the peritoneal fluid of women with endometriosis may cause endometriosis-related pain. Inflammatory cytokine synthesis appears greater in those with peritoneal endometriosis compared to endometrioma. This may explain the relationship between dyspareunia and endometriosis. Initial clinical observations in the studied patients show a significant reduction in pain symptoms caused by endometriosis, including dyspareunia, after administration of dienogest (a birth control medication that combines estrogen and progestin). There is a need for an in-depth analysis of endometriosis in patients without other factors related to dyspareunia. In the case of peritoneal endometriosis, patients may be asked to reduce their frequency of sexual activity until the endometriosis resolves to avoid the development of secondary, psychogenic vaginismus. Treatment that combines physiotherapy and dienogest may be advisable for these patients. Other non-medical endometriosis treatments include: (1) antioxidants [23]; (2) Chinese herbal medicines [24,25]; (3) acupuncture [26]; (4) manual therapy [27]; (5) reflexology; (6) homeopathy; (7) psychotherapy; (8) exercise and sports; and (9) nerve blocks such as a superior hypogastric plexus block [28].

## 5. Physiotherapy as Treatment Support in Endometriosis

Endometriosis treatment is complex because of the presence of chronic pelvic pain. Practitioners must assess pain intensity and look for symptoms suggestive of endometriosis. The main symptoms that suggest endometriosis are severe dysmenorrhea, deep dyspareunia, painful bowel movements during menstruation, urinary symptoms, and infertility [29]. Endometriosis is often associated with diffuse pelvic pain, which affects quality of life. Pelvic organ pain is often perceived as somatic pain due to innervation from the same level, which is related to peripheral sensitization [30,31,32,33,34,35].

Gynecological diseases can cause pain in the pelvic area and in the lower spine. If the examination excludes musculoskeletal causes, reproductive system dysfunction should be considered. This requires obtaining a more detailed medical history with a specific focus on the genital organs.

For example, back pain can be caused by pregnancy, ovarian cysts, uterine retroversion, endometriosis, uterine fibroids, or upper genital tract inflammation [35]. Increased abdominal cavity tension caused by hypertonia of the pelvic urogenital diaphragm can shorten the iliopsoas muscle and adversely affect the lower spine. Disease processes in this area manifest as increased protective tissue tension. Other causes of pelvic pain include uterine misalignment, (ectopic) pregnancy, uterine fibroids, ovarian cysts, intrauterine device (IUD) use, endometriosis, pelvic prolapse, vulvodynia, premenstrual syndrome, adhesions, polyps, and varicose veins. Sacrum and sacroiliac joint pain can be caused by neoplasms of the reproductive system, uterine prolapse, pelvic inflammatory disease, uterine retroversion, pregnancy, ovarian cysts, IUD use, or endometriosis. Pain in the lower spine, pelvis, sacrum, and sacroiliac joint may also be caused by sexual abuse [32].

Myofascial pain is a manifestation of dysfunction in the muscle and the surrounding myofascial/connective tissue. Simons et al. found the lifetime prevalence of myofascial pain to be 85% in the general population [33,34]. Myofascial trigger points (MTrPs) are small, palpable, hypersensitive nodules located on tense skeletal muscles in an area of sustained contraction. MTrPs may be active or latent and can occur in the vagina, rectum, urethra, pubic bone, vagina, coccyx, abdomen, lower back, and back of the thighs [36].

Myofascial pelvic pain is frequently characterized by dyspareunia, painful urination or *dysuria*, and dyschezia; however, these symptoms may be manifestations of other non-pelvic floor problems or problems with the pelvic viscera [36]. Pelvic pain after sexual activity may persist for up to three days and exert profoundly adverse effects on sexual health [37]. Regular pelvic floor exercises increase blood flow and stimulate pelvic floor proprioception, both of which may contribute to a more intense orgasm. Pelvic floor physiotherapy is safe and effective and can significantly improve symptoms associated with pelvic dysfunction. Using stretching techniques for all muscles related to the pelvis, abdomen, back, upper legs, and nerve structure mobilization is necessary to improve pelvic area function. Stretching exercises and strength training restore balance and stability, lengthen pelvic floor muscles, lengthen fascial tissue, and reduce nervous tension [38]. Such exercises are often recommended for patients with endometriosis; the addition of manual therapy such as visceral therapy may help further reduce pain and improve quality of life.

Somatic dysfunctions can lead to disproportionate pressure gradients in the pelvic cylinder, potentially resulting in stasis, inflammation, retention, visceral disorders, and vasomotor restrictions [39,40,41].

Manual therapies, such as visceral treatments, for reproductive system dysfunction involve restoring postural balance, breathing, pelvic activity, and balancing the pressures between particular diaphragms in the body [33]. In the anterior projection, the internal organs tend to descend through the inspiratory (flattened) position of the diaphragm. Anterior projection is also characterized by extensive trunk tension, resulting in an inappropriate pressure gradient and a forward pelvic tilt. Posterior projection features the expiratory (domed) position of the diaphragm, a backward pelvic tilt, and tension within the sacroiliac joints and the cervicothoracic junction [35].

Uterine palpation can determine the presence of adhesions, shape, mobility, and location of the uterus. The physiological condition of the uterus can be comprehensively assessed when there is no strong resistance or pain in the lower abdominal area. The uterus should be somewhat flexible. A strong reaction to pressure may signal adhesions. Completely absent resistance may indicate uterine retroversion [40,41].

Uterine relaxation techniques are useful for treating reduced uterine mobility. Additionally, these techniques support vascular drainage of the uterus [40,41].

For patients complaining of painful intercourse, premenstrual syndrome, and painful periods, another treatment seeks to restore uterine broad ligament mobility [42,43].

## 6. Conclusions

The COVID-19 pandemic forced patients to seek alternatives to surgery to relieve pain. Our experience clearly shows the benefits of visceral therapy. Importantly, quality of life should be carefully assessed in patients with symptomatic endometriosis [29]. This can be accomplished using the abbreviated generic Quality of Life Scale developed through the World Health Organization (WHOQOL-BREF) [44]. Prospective trials that compare surgical and visceral therapy are required to improve our ability to manage pain in patients with various forms of endometriosis.

## Data Availability

Not applicable.

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
