# Peer review of "The Role of Visceral Therapy in the Sexual Health of Women with Endometriosis during the COVID-19 Pandemic: A Literature Review"

_jcm, 2022, doi:10.3390/jcm11195825_

Round 1
Reviewer 1 Report (New Reviewer)
We congratulate the authors about their assessment of sexuality of endometriosis patients during covid pandemic
we would like to suggest the following points
line 23 replace females by patients
line 29 replace occurrence by presence
line 51 the author did not develop history taking in endometriosis diagnosis and emphasize more physical exam
line 71 replace ovulation inhibition by hormonal
line 88 according to ESHRE , laparoscopy is not the gold standard for diagnostisis can the author develop?
line 173 what do the author mean by this statement
Author Response
Manuscript JCM-1900540 entitled “Endometriosis and sexual health—gynecological and physio-therapy considerations during the COVID-19 pandemic and literature review”
Dear Prof. Dr. Sylvia Mechsner
Special Issue Editor
Special Issue "Endometriosis: Current Perspectives on Diagnosis and Treatment"
A special issue of Journal of Clinical Medicine (ISSN 2077-0383)
The response to the Reviewer 1 comments are given below. Changes in the manuscript have been made in accordance with the suggestions of the Reviewer 1 and all changes have been underlined in the manuscript.
- line 23 replace females by patients
Replay: Thank you for the comment. We changed it.
- line 29 replace occurrence by presence
Replay: Thank you for the comment. We change it.
- line 51 the author did not develop history taking in endometriosis diagnosis and emphasize more physical exam
Replay: Thank you for the comment. We extended and emphasized more physical exam. History taking is described in detail two paragraphs later. We also added information that to date no study has assessed whether use of questionnaires or the use of symptom diaries compared to traditional history taking techniques has shortened or improved the diag-nosis of endometriosis neither for screening nor for triaging of symptomatic patients.
- line 71 replace ovulation inhibition by hormonal
Replay: Thank you for the comment. We changed it.
- line 88 according to ESHRE , laparoscopy is not the gold standard for diagnostisis can the author develop?
Replay: Thank you for the comment. Laparoscopy was gold standard for diagnosis of endometriosis, however, according to ESHRE 2022 guideline, typical clinical symptoms, physical exam and advances in the quality and availability of imaging modalities for at least some forms of endometriosis may reliably detect or exclude endometriosis. We changed and develop this paragraph according to your suggestion.
- line 173 what do the author mean by this statement
Replay: Thank you for the comment. We added: “nerve blocks such as superior hypogastric plexus block”
I would like to thank the editors and reviewer for their detailed understanding and analysis of the contents of this paper as well as for their constructive comments.
Sincerely,
Tomasz Goździewicz, M.D., Ph.D.
Division of Gynecology, Department of Perinatology and Gynecology, Poznan University of Medical Sciences, Poland
Reviewer 2 Report (New Reviewer)
Thank you for allowing me to review this article. The authors have conducted a scoping review of the literature to analyse the relationship of endometriosis and sexual health, with specific emphasis on the gynaecological and physiotherapy considerations during the Covid-19 pandemic.
The authors have provided a brief resume of the etiology, manifestations and possible medical strategies for treating the endometriosis, although their emphasis is to discuss management of dyspareunia due to endometriosis whilst adding Visceral therapy as a option.
(1) I sense this article has as its primary intention to advocate the use of Visceral therapy in managing sexual health in women suffering with endometriosis especially during times when the women have constraints to surgical treatment, namely pandemics. The title is misleading or inappropriate.
(2) The authors should briefly state why they undertook a scoping analysis vs a systemic analysis.
(3) The authors should primarily discuss sexual health and not all the other possible causes of pain or complications of endometriosis. The title should dictate intention and content.
(4) Discussing Visceral therapy as a complementary therapeutic option requires a greater scientific appraisal than what has been provided.
(5) This treatment option should be supported only as a complementary strategy and should only include gynaecological pelvic physical therapy. The other types of visceral therapy may have general benefits, but do not have major support for dysparunia. The authors must include information pertaining to neurologic pathways and dyspareunia and specifically with view to the pudendal nerve.
(6) There are data to support that Visceral therapy, from a diagnostic and efficacy point of view. does not improve clinical endpoints in conditions associated with anatomical and physiological aberrations. The authors need to expand on this and include it in their studies.One particular review analysed 8 diagnostic studies and 6 efficacy studies and found no significant differences in clinical outcomes
(7) May well be worthwhile to change the title so as to emphasise the complementary role of Visceral therapy and sexual health in women with endometriosis. Then only address sexual health and Visceral therapy in the text and delete all the other information not pertaining to the topic.
Author Response
Manuscript JCM-1900540 entitled “Endometriosis and sexual health—gynecological and physio-therapy considerations during the COVID-19 pandemic and literature review”
Dear Prof. Dr. Sylvia Mechsner
Special Issue Editor
Special Issue "Endometriosis: Current Perspectives on Diagnosis and Treatment"
A special issue of Journal of Clinical Medicine (ISSN 2077-0383)
The response to the Reviewer 2 comments are given below. Changes in the manuscript have been made in accordance with the suggestions of the Reviewer 2 and all changes have been underlined in the manuscript.
- I sense this article has as its primary intention to advocate the use of Visceral therapy in managing sexual health in women suffering with endometriosis especially during times when the women have constraints to surgical treatment, namely pandemics. The title is misleading or inappropriate.
Replay: Thank you for the comment. We changed the title of manuscript to: “Role of visceral therapy and sexual health in women with endometriosis during the COVID-19 pandemic - literature review.”
- The authors should briefly state why they undertook a scoping analysis vs a systemic analysis.
Replay: Thank you for the comment. We undertook a scoping analysis vs a systemic one because there are limited data about visceral therapy in endometriosis. This review is introduction to our prospective study concerning visceral therapy in endometriosis.
- The authors should primarily discuss sexual health and not all the other possible causes of pain or complications of endometriosis. The title should dictate intention and content.
Replay: Thank you for your comment. It is known that awareness about endometriosis among medical professionals are still not satisfactory , so our idea was to briefly review most important facts about etiology, diagnosis and management of endometriosis and later focus on topic itself.
- Discussing Visceral therapy as a complementary therapeutic option requires a greater scientific appraisal than what has been provided.
- This treatment option should be supported only as a complementary strategy and should only include gynaecological pelvic physical therapy. The other types of visceral therapy may have general benefits, but do not have major support for dysparunia. The authors must include information pertaining to neurologic pathways and dyspareunia and specifically with view to the pudendal nerve.
Replay: Thank you for your comment. We modified and changed name of that part of manuscript as “Physiotherapy as treatment support in endometriosis”
- There are data to support that Visceral therapy, from a diagnostic and efficacy point of view. does not improve clinical endpoints in conditions associated with anatomical and physiological aberrations. The authors need to expand on this and include it in their studies.One particular review analysed 8 diagnostic studies and 6 efficacy studies and found no significant differences in clinical outcomes
Replay: Thank you for your comment. We agree with “Reliability of diagnosis and clinical efficacy of visceral osteopathy: a systematic review” Albin Guillaud at all manuscript. This review showed high risk of bias in results of most studies. Only two studies had low risk of bias and do not support the efficacy of visceral techniques in low back pain and for very low birth weight infants. However, in this review there were no strict data about endometriosis and visceral therapy. As we mentioned previously, this review is introduction to our prospective study results concerning visceral therapy in endometriosis.
- May well be worthwhile to change the title so as to emphasise the complementary role of Visceral therapy and sexual health in women with endometriosis. Then only address sexual health and Visceral therapy in the text and delete all the other information not pertaining to the topic.
Replay: Thank you for the comment. We changed the title of manuscript to: “Role of visceral therapy and sexual health in women with endometriosis during the COVID-19 pandemic - literature review.”
I would like to thank the editors and reviewer for their detailed understanding and analysis of the contents of this paper as well as for their constructive comments.
Sincerely,
Tomasz Goździewicz, M.D., Ph.D.
Division of Gynecology, Department of Perinatology and Gynecology, Poznan University of Medical Sciences, Poland
Round 2
Reviewer 1 Report (New Reviewer)
N/A
Reviewer 2 Report (New Reviewer)
Major revision has been undertaken by the authors which has improved the article substantially
This manuscript is a resubmission of an earlier submission. The following is a list of the peer review reports and author responses from that submission.
Round 1
Reviewer 1 Report
1) Needs significant English corrections to be published.
2) Is mainly a review of endometriosis. Limited science. Limited interventions discussed. Goal of paper is to address COVID limitations and the impact on endometriosis treatment - I did not get a sense of that at all during reading. At the end, the authors mention interventions but these still require in person therapy or treatment, which is limited by COVID. Mention a questionnaire - ref 42 in the conclusion, yet there is no description of it anywhere else.
3) Consider designing this paper as strictly a review or rewrite it completely to demonstrate specific barriers during Covid, specific interventions you did, and your outcomes.
Reviewer 2 Report
thank you giving me the chance to review this paper hilgighteing the role of physiotherapy for endometriosis, in particular after COVID 19 outbreaks.
Despite good overall merit I have some comments:
- please discuss abour rare but potential localization of extrapelvic endometriosis (eg doi 10.1016/j.jmig.2012.03.005)
- please discuss about the potential use of transperineal ultrasound to detect myofascial syndrome (eg doi 10.1007/s00192-019-03963-4)
- lines 163-165: In the case of peritoneal endometriosis, a recommendation may be made to temporarily reduce the frequency of sexual contact until the endometriosis has healed to avoid the development of secondary, psychogenic vaginismus. Please provide references for this sentence.
Reviewer 3 Report
The authors reviewed the association between endometriosis and sexual health during the pandemic of SARS-CoV-2. This theme is important; however, the reviewer feels that the SARS-CoV-2 situation is improving in many countries. My suggestions to improve the manuscript are as follows.
Several sentences are difficult to read. I recommend that the manuscript be reviewed by a person with professional proficiency in English to correct errors in grammar, punctuation, word choice, and sentence construction to improve the flow of ideas expressed in the article to ensure that the document reads as though written by a native English speaker.
Introduction
Prior studies supporting the authors' statements should be cited.
Line 20-23
Especially, the authors need to cite previous studies that support authors' statements. In my country, medical treatment and surgeries for endometriosis are not postponed.
Lines 91-170
How does SARS-CoV-2 affect Dyspareunia and sexual health?
Lines 171-327
How does SARS-CoV-2 affect the visceral therapy in endometriosis?
Lines 233-283
This is a run-on paragraph and is excessively lengthy. Long winding sentences tend to confuse readers and may lead to misinterpretation. Short sentences are preferred for better clarity and readability. I suggest revising this segment using shorter sentences for easier readability.
While the authors focus on the medical treatment for endometriosis during SARS-CoV-2, this review is superficial.